# Compatibility between "Arbequina" and "Souri" Olive Cultivars May Increase Souri Fruit Set [†]

**Iris Biton [1], Yair Many [1], Ali Mazen [2] and Giora Ben-Ari [1,\*]**

[1]  Institute of Plant Science Volcani Center, ARO, HaMaccabim Road 68, P.O.B 15159 Rishon LeZion 7528809, Israel; ivrb28@agri.gov.il (I.B.); yairm@agri.gov.il (Y.M.)
[2]  The Israeli Olive Oil Council, Deir Hanna, 2497300, Israel; dr.mazn_ali@hotmail.com
\*  Correspondence: giora@agri.gov.il; Tel.: +972-3-9683922
†  Compatibility of 'Souri' olive cultivar.

**Abstract:** The "Souri" olive cultivar, which is autochthonous to the eastern Mediterranean region, has been the major olive variety cultivated traditionally under rain-fed conditions in northern Israel. The aim of this study was to determine the optimal pollen donor for the olive cultivar Souri in order to maximize Souri fruit set. Artificial cross pollination of Souri flowers with several local varieties has identified the "Nabali" as the most efficient pollinizer of the Souri. However, further experiments using artificial cross pollination conducted with cultivars not common to this region have revealed the "Arbequina" as a more efficient pollinizer of the Souri cultivar than the Nabali. Based on a preliminary paternity analysis, the Nabali was identified as the dominant pollinizer of Souri trees in traditional olive orchards in the north of Israel. However, in a multi-variety orchard, molecular paternity analysis has shown Arbequina to be the most frequent pollinizer. We then tested, during two consecutive years, whether the presence of a pollen-producing Arbequina tree adjoining Souri trees in the field will increase their fruit set. We found that Souri fruit set was 8.36% when pollinized by an Arbequina tree in close proximity to them, significantly higher than the fruit set of 5.6% for Souri trees without the nearby Arbequina cultivar. On the basis of these trials, we expect that the yield of Souri orchards will improve if Arbequina trees are planted.

**Keywords:** *Olea europaea*; pollination; pollen donor; SNPs; fruit production

## 1. Introduction

Fruit production in olive orchards is dependent to a large degree on successful fertilization. Numerous endogenous and exogenous factors determine fruit production [1]. Environmental factors such as rain, wind, and temperature, can greatly affect flowering and fertilization [2]. Rising temperatures and drought resulting from climate change may negatively affect flowering, causing abscission of flowers, reducing fertilization and thus decreasing fruit set [2]. Clearly, maximizing fruit set potential is of great importance in raising yields of local olive groves. The olive (*O. europaea*) has a homomorphic sporophytic diallelic self-incompatibility system [3]; thus, most olive cultivars are self-incompatible while fertilization efficiency varies between cultivars [4,5].

### 1.1. Souri Cultivar in Israel

The Souri cultivar has been cultivated traditionally under rain-fed conditions in Israel for hundreds of years. Even today this cultivar occupies about two thirds of the olive groves in the country [6]. The Souri olives in Israel are grown mainly in single-variety orchards by traditional farmers in the north of Israel. In most areas in the north of Israel, the Souri cultivar is cultivated exclusively. However, isolated trees of other cultivars, such as "Zakari", Nabali, "Chimlali", and "Amrahani" are sporadically

distributed throughout the area, within the Souri orchards. The Souri cultivar is characterized by its high adaptability to semi-arid conditions and occasional droughts, shallow and stony marginal soils, and varying climatic conditions [7,8]. Its fruit has a high aromatic oil content reaching more than 30% by commercial extraction [9]. The Souri is a dual-purpose variety, and in addition to being the basis of the local oil producing industry, it also provides green fermented table olives. Thus, Souri orchards were traditionally planted for generations around mountain villages in the northern part of the country. With the introduction of intensive cultivation and irrigation to the newly developing commercial olive oil industry [10,11], it became apparent that the Souri cultivar is not ideally suited to irrigation agriculture. This is probably due to generations of selection of the Souri for the astringent growth conditions of local traditional agriculture.

### 1.2. Compatibility between Olive Cultivars

The olive (*Olea europaea* L.) is a wind-pollinated, preferentially allogamous species. Although olive flowers are visited by insects, wind remains the main factor in pollination of olives [1,12]. Long-distance wind dispersal of olive pollen grains up to 12 km has been observed under optimal conditions [13]. However, efficient cross-fertilization between compatible cultivars has been shown to take place at distances of up to 250 m away [14].

The olive is characterized by a homomorphic sporophytic diallelic self-incompatibility (DSI) system. The olive S-locus consist of two alleles, *S2* (dominant) and *S1* (recessive), which produce two incompatibility groups, *S1S2* corresponding to one self-incompatibility group and *S1S1* to the other [3]. This system prevents self-fertilization and regulates compatibility between cultivars, so that cultivars bearing the same incompatibility group are incompatible. Despite this system, some varieties have been found to be self-compatible. This is explained by a mechanism known as pseudo-self-compatibility [15]. Efficient pollination and fertilization depend on many factors, such as the duration of stigma receptivity, the number of pollen grains, pollen–ovule ratio, stigma morphology, but most importantly, the presence of pollen from a compatible cultivar [3,14–18]. Some studies have found varying efficiencies of pollination between compatible cultivars and certain cultivars have been shown to pollinate specific cultivars more efficiently than others [4,19–21].

DNA markers offer a reliable way to test the paternity of seeds by comparing the mother plant genotype to the genotype of the embryo in order to identify the pollen donor among several candidates. Microsatellite markers have been widely used for paternity analysis in olives [4,20–22]. In recent years, the use of SNP (single nucleotide polymorphism) markers in olives has been found useful for this purpose [23–25]. The main advantage of microsatellite use is its very high polymorphism. This enables discrimination between individuals by only few microsatellite loci. However, microsatellite markers have higher mutation rates and they are less reproducible compared to SNPs [4,6,24,26,27]. Although SNPs are diallelic (compared to the highly polymorphic microsatellites), the use of SNPs instead of microsatellite markers has several advantages. SNPs are more abundant, genotype analysis can be performed at high-throughput scales and SNP alleles contain different nucleotides, which are easy to characterize and enable high reproducibility.

The aim of this study was to identify the optimal pollinizer of the Souri cultivar in order to increases its productivity. We used artificial cross-pollination as well as paternity analysis, followed by a successful cross pollination field trial with an effective pollen donor cultivar, in order to improve Souri fruit set percentage.

## 2. Material and Methods

### 2.1. Characterization of the Flowering Period

We characterized the flowering period of the olive cultivars Souri, "Barnea", "Picual", "Picholine Languedoc", "Manzanillo", Arbequina, "Muhasan", "Koroneiki", "Leccino", "Coratina", Nabali, and "Frantoio". Characterization of the flowering period of each cultivar was performed during the spring

of 2016 in the Israeli germplasm collection located at the Volcani Center (ARO) in Rishon LeZion, Israel (31°58′57.8″ N 34°49′47.2″ E). The collection consists of one hundred nineteen 22 year-old, irrigated olive tree cultivars spaced 5 × 6.5 m apart, [24]. Six branches with about 100 flower buds each, from each tested cultivar were marked, and the actual number of buds on each branch was recorded. The trees were observed at regular intervals of 3–4 days from the beginning of March until the end of May. At each observation, the number of closed buds, open flowers and dried flowers was recorded. For each cultivar, the flowering period was characterized as the period beginning the day 10% open flowers were observed and lasting until 90% of the flowers were dry.

*2.2. Artificial Pollination*

2.2.1. Self-Compatibility Examination

The self-compatibility examination was carried out during the spring of 2016, by artificial pollination of 5 olive trees (cv Souri) selected from the Israeli germplasm collection. As a control, we used the Barnea cultivar as a pollen donor, since we already found that Barnea and Souri are compatible [4]. Approximately 10,000 closed flowers on 40 branches (8 branches per tree) were counted. Branches were covered by paper bags and at commencement of the flowering period, 10 open Barnea inflorescences were inserted in each of 20 bags chosen at random from the 40 covered branches. The other 20 branched served as a self-pollinated control. Paper bags were shaken every day and removed at the end of the Souri flowering period (May 2016). Since fruitlet drop occurs during the 30 days after anthesis [28], fruits were counted two months after anthesis (July 2016), and fruit set was calculated.

2.2.2. Artificial Cross-Pollination

During the spring seasons of 2016, 2017, and 2018, artificial cross-pollination was carried out in three 30 years old, non-irrigated Souri orchards near the village of Deir Hanna in the north of Israel (S1–32°52′02.2″ N 35°21′40.7″ E, S2–32°52′15.3″ N 35°22′11.1″ E and S3–32°52′30.0″ N 35°22′10.4″ E). All surrounding olive groves covering the landscape can be considered mono-cultivar orchards of Souri. In each of the three years of artificial cross-pollination, for each donor cultivar tested, in each orchard, 4 Souri branches from each of 5 trees, each branch containing at least 250 flowers before opening, were enclosed in paper bags; a total of 15,000 flowers for each tested pollinizer in each year. After the enclosed Souri flowers opened, 5 pollinizer inflorescences from the tested donor were inserted into each bag and after two months, the number of developing fruits was determined. As a free pollinated control, we used the same number of branches, flowers were counted, and the branches left uncovered.

In 2016, Nabali, Zakari, and Muhasan trees, which are distributed sporadically throughout the area of Souri orchards in the north of Israel, were the cultivars chosen as candidate pollinizers of the Souri, and self and free pollinated branches served as controls. In 2016, 20 Souri branches from each tree were chosen, four branches for each pollen donor. In 2017 the candidate pollen donating cultivars were Nabali and Koroneiki and in 2018 Nabali and Arbequina. In both years (2017 and 2018) the fruit set after free pollination was also monitored. In 2017, we tested the Koroneiki instead of the Barnea as the Souri pollinizer, since we had a technical problem collecting Barnea pollen at the Volcani Institute. Fortunately we had many Koroneiki flowers available, which were also known to be compatible with Souri [4]. In 2018, we choose Arbequina as our pollen donor, based on the results of paternity analysis. Artificial cross-pollination using paper bags affects the microclimate around the flowers and can cause damage if left on the branch for a long period. The relatively long distance from our base at the Volcani Institute to the site of our cross-pollination trials near the village of Deir Hanna caused technical difficulties which resulted in a delay in removing the paper bags covering the flowers during the seasons of 2017 and 2018. As a result, the Souri fruit set of artificially cross-pollinated flowers was low. Therefore, we normalized the fruit set data of the various years based on the fruit set of free and Nabali cross pollination, which was performed in all three years.

### 2.3. Paternity Analysis

For molecular paternity analysis, one olive fruit was collected from each of 7–15 Souri trees (depending on the orchard) in five commercial olive orchards in the north of Israel. Among the five orchards, four were mono-cultivar traditional non-irrigated orchards (Zemer, Yasif, Rame, and Deir Hanna), whereas the fifth was a multi-cultivar irrigated grove in Gshur, consisting of 12 cultivars; each row in the orchard containing a different cultivar. DNA was extracted from leaves sampled from the Souri trees in each orchard as well as the embryos extracted from each sampled fruit. In order to identify the pollen donor, DNA samples were genotyped using the 96 most informative SNPs, selected from the 138 SNPs used in an earlier study to characterize our germplasm collection [24].

DNA Extraction and Genotyping

Genomic DNA was extracted from leaves using the CTAB method [29] and from embryos using high-throughput DNA extraction method [30]. Large-scale genotyping of SNPs was performed on a Fluidigm 96.96 Dynamic Array using the genotyping EP1 System (San Francisco, CA, USA). Fluorescence intensity was measured with the EP1 (Fluidigm Corp, San Francisco, CA, USA) reader and plotted on two axes. Genotypic calls were made using the Fluidigm SNP Genotyping Analysis program. Souri embryos from plants in pots were characterized using five SSR markers as described earlier [4].

### 2.4. Increasing Fruit Set by Cross Pollination with Arbequina

Cross pollination with Arbequina was performed with five-year old Souri trees in 50 L pots. The trees were irrigated and kept at the Volcani center until the spring of 2018 and 2019, when they were moved to the experimental groves and their fruit set was monitored. At the end of each spring, the Souri trees were returned to the Volcani center. In the spring of 2018 and 2019, the Souri trees were separated into two groups of 4 each and placed in two locations in non-irrigated Souri groves near the village of Deir Hanna in the north of Israel (S1–32°52′02.2″ N 35°21′40.7″ E and S3–32°52′30.0″ N 35°22′10.4″ E). The two locations were 1.2 km one from the other. The surrounding landscape is covered by exclusively Souri mono-cultivar olive orchards, with several Nabali, Zakari, and Muhasan trees sporadically distributed throughout the area, within the Souri orchards. Near one of the groups of Souri trees (S3), one five years old Arbequina tree was placed in a pot (Supplementary Figure S1). All the pots were irrigated by hand once every two days for the period of the experiment (spring 2018 and 2019). In each of the two years, the number of flowers was determined at the end of March, before the flowers opened. In mid-June, fruit number was determined, and plants were returned to the Volcani center.

### 2.5. Statistical Analysis

Fruit set of Souri as a result of artificial cross-pollination with various pollen donor was characterized using randomized block design, whereas the three Souri plots served as random blocks. However, since there were no differences between the three plots ($F_2 = 0.0128$, $p = 0.987$), one-way analysis of variance (ANOVA), with the pollen donor as the independent variable, was used for the analysis of fruit set. When fruit set between years was also compared, pollen donor and year (random) served as independent variables, fruit set served as a dependent variable and a full factorial two-way ANOVA analysis was performed. When we encountered significant factors, a Tukey–Kramer test was performed in order to rank the various levels. Free pollination resulted in different fruit sets in 2016, 2017, and 2018. Souri fruit set after artificial cross pollination with Nabali in 2017 and 2018 was significantly lower than in 2016. We then used the fruit set data of free pollination as well as the data of the cross-pollination with Nabali during the three years of cross pollination experiments as a base to analyze the effect of the pollen donor, the year, and the interaction between the two factors on fruit set by a full factorial two-way ANOVA analysis. We normalized the Souri fruit set over the three years of

the cross-pollination experiments independently, for each pollen donor. All statistical analyses were performed using JMP software [31].

## 3. Results

### 3.1. Flowering Period

The flowering periods of the Souri cultivar as well as ten potential pollen donors, were recorded during 2016. The first cultivar to bloom was the Barnea at day 86 and the last was Leccino at day 108. The Souri bloomed for two and half weeks, from day 93 till day 110. We found that the flowering period of all 10 potential pollen donors overlapped to some extent with the Souri cultivar (Figure 1).

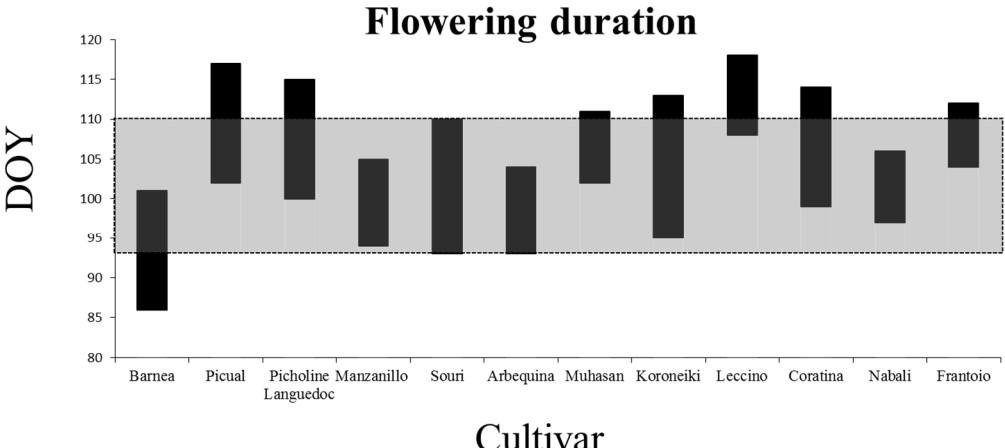

**Figure 1.** Flowering periods of 12 cultivars during 2016. Bars indicate the flowering period from its beginning (10% of opening flowers) to its end (90% of dried flowers). The Y axis represent the day of year (DOY). The gray rectangle represents the Souri flowering period.

### 3.2. Artificial Pollination

In order to exam the self-compatibility characteristic of the Souri cultivar, we compared artificial self-pollination to cross-pollination with Barnea at the Volcani center in 2016. We found the Souri fruit set percentage to be 1.5 when Barnea served as a pollen donor, whereas no fruit set appeared when self-pollination was attempted (ANOVA, $F_1 = 44.8$; $p = 1.029 \times 10^{-8}$). All other artificial pollination trials were performed at Deir Hanna during the seasons of 2016–2018.

Analyzing the fruit set of the various pollen donor among the three years of experiment showed significant interaction between the pollen donor and the year ($F_2 = 26.8$; $p = 6.64 \times 10^{-11}$). Therefore, we normalized the Souri fruit set over the three years of the cross-pollination experiments. In 2016 we tested the self, free, and cross compatibility of Souri with several cultivars. We observed significant differences in rates of fertilization ($F_4 = 18.1$; $p = 3.65 \times 10^{-14}$), depending on the identity of the pollen donor (Figure 2). Free pollination gave a fruit set of 4.2%. Cross-pollination with Nabali resulted in a fruit set of 3.9%, which was not statistically different from free pollination. Cross-pollination with Zakari and Muhasan resulted in a lower fruit set compared to the results with the Nabali cultivar—2.5% and 2.4%, respectively. The small number of fruit sets after self-pollination (0.4%), was probably a result of contaminated pollen. In 2017 and in 2018 we performed cross pollination of Souri flowers with Nabali pollen, which was found to be the most efficient pollen donor in 2016. These trials included also cross-pollination with Koroneiki in 2017, and with Arbequina in 2018. We also tested free pollination in both years. The Souri fruit set of freely pollinated flowers was 4.2%, and Souri fruit set of flowers cross-pollinated by Nabali was similar (3.9%). However, the Souri fruit set of flowers cross pollinated with Koroneiki (5.97%) in 2017 and with Arbequina (6.9%) in 2018 was significantly higher ($F_2 = 7.3$; $p = 9.88 \times 10^{-4}$ and $F_2 = 11.8$; $p = 7.97 \times 10^{-5}$ in 2017 and 2018, respectively).

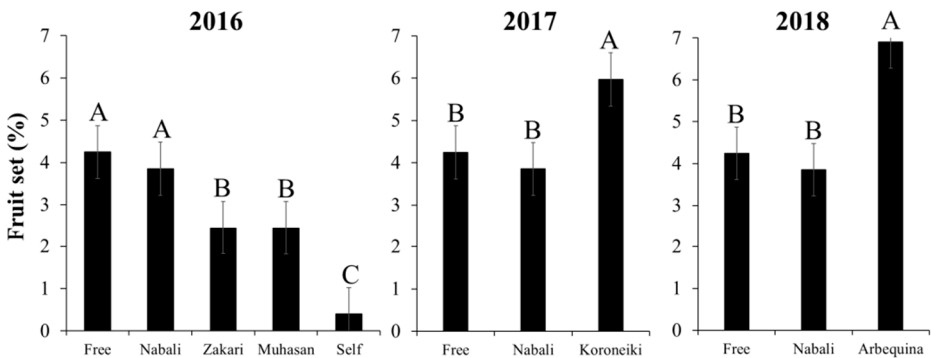

**Figure 2.** Average fruit set of Souri after cross-pollination with various cultivars in the years 2016–2018. For comparison we measured the fruit set under a free pollination strategy. Letters above the columns represent significant differences obtained in the Tukey–Kramer test ($p < 0.05$). Error bars represent confidence intervals ($p < 0.05$).

*3.3. Paternity Analysis*

Based on 96 SNPs markers, we analyzed the pollen donor of 52 Souri fruits per year, sampled from five different olive orchards—four traditional mono-cultivar groves and one multi-cultivar orchard during October of 2016 and 2017. We sampled 33% of the fruits from the multi-cultivar orchard in Gshur, 21% from the mono-cultivar orchard in Deir Hanna, and 46% of the fruits were from the mono-cultivar orchards in Zemer, Yasif, and Rame. Due to suspected interference by unknown pollen donors and some technical problems in the PCR reaction, probably due to unsatisfactory DNA quality, the pollen donors of only 67% fruits in 2016 and 77% in 2017 were identified. The number of embryos with an unidentified pollen donor was 0, 7, 3, 4, and 3 in 2016 and 2, 0, 2, 3 and 5 in 2017, whereas the number of identified pollen donor was 18, 4, 3, 5, and 5 in 2016 and 14, 11, 7, 6, and 2 in 2017, in Gshur, Deir Hanna, Zemer, Yasif and Rame, respectively. In order to explain the high number of embryos with unidentified pollen donors in the mono-cultivar orchards, we sampled six non-Souri trees distributed within the Souri mono-cultivar orchards. We determined that none of them was the pollen donor of any of the analyzed embryos and that bore a different genotype.

Figure 3 represents the pollen donors of all analyzed fruits in 2016 and 2017. In 2016, the Nabali cultivar was identified as the most frequent pollen donor in all four traditional mono-cultivar orchards. In 2017, although the Nabali was not identified as a pollen donor in any of the sampled fruits in the mono-cultivar orchards in Yasif and Rame, it was the main pollen donor in the mono-cultivar orchards in Deir Hanna and Zemer. In total, Nabali was identified as the pollen donor of 62.5% of the fruits sampled in the mono-cultivar orchards in 2016 and in 38% of the fruits sampled in 2017, the most frequent donor in both years. In the multi -cultivar orchard in Gshur, Arbequina was the most frequent pollen donor in both years, and Barnea second.

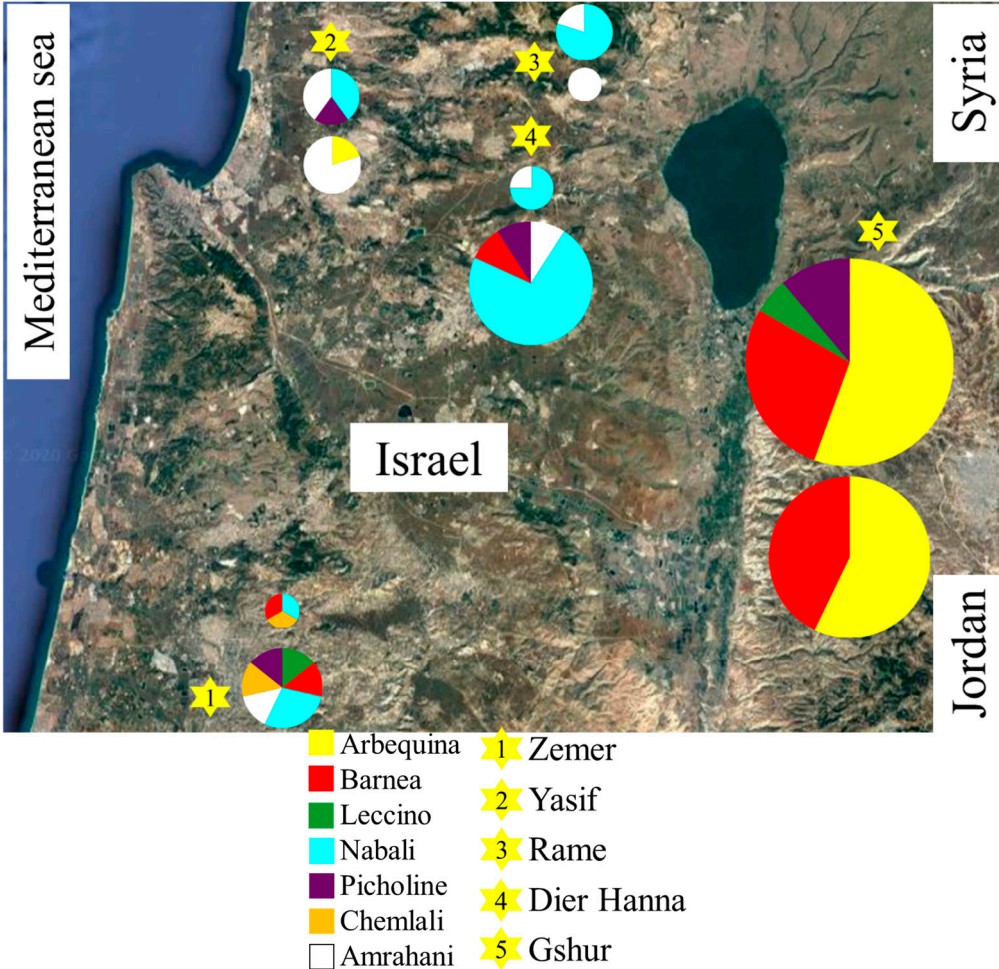

**Figure 3.** Paternity analysis of Souri embryos sampled from various orchards demonstrating the dominance of Nabali as pollen donor in the four traditional mono-cultivar Souri orchards and of Arbequina in the multi-cultivar orchard. Location (by number) of the sampled orchards appears in yellow stars at each location. Two pie charts of 2016 (upper pie) and 2017 (lower pie) adjacent to each star, present the results of pollen donor identity. Each pollen donor is represented by a different color. The size of the pie is proportional to the number of embryos analyzed.

## 3.4. Increasing Fruit Set by a Compatible Cultivar

Since the Arbequina was found to be the dominant pollinizer in the multi-cultivar orchard, we then tested its ability to increase fruit set of Souri trees in the mono-cultivar orchards, by a field cross pollination trial with Arbequina. In order to avoid differences between orchards, we used five years old Souri trees growing in pots. We divided the Souri trees into two groups. The first group was placed in an orchard in Deir Hanna and the second group was located in a different orchard near the same village, but 1.2 km away. To the second group we added a five-year old Arbequina tree in a pot. The potted trees were kept in the Volcani center all year and transferred to the orchards in Deir Hanna for the flowering and fruit set period. We performed this experiment in the spring of 2018 and again in 2019. In 2018, the group of Souri trees without an Arbequina pollen donor contained a total of 13,697 flowers, of which, 605 were set to fruits (4.4%). The group of Souri trees with an added Arbequina tree nearby contained a total of 10,659 flowers, of which, 888 of them set to fruits (8.3%). In 2019, the group of Souri trees without Arbequina contained a total of 9370 flowers, of which 543 set to fruits (5.8%). The group of Souri plants with Arbequina nearby contained a total of 9010 flowers, of which 753 were set to fruits (8.4%). Two-way ANOVA testing revealed that the effect of the presence of Arbequina nearby was significant ($F_1$ = 72.4; $p$ = 0.013), whereas the effect of the year ($F_1$ = 3.4;

$p = 0.578$), and the interaction between the year and the presence of Arbequina ($F_1 = 3.2$; $p = 0.592$), were not significant. Therefore, we analyzed the set of data from both years together. In both years, the group of Souri plants without Arbequina contained a total of 23,067 flowers of which 1148 were set to fruits (5%). The group of Souri trees with Arbequina nearby, had a significantly higher fruit set ($p = 0.0023$) and contained a total of 19,669 flowers in which 1641 of them were set to fruits (8.3%) (Figure 4). To ensure that the Arbequina pollen was responsible for the Souri high fruit set, in October 2018 we sampled four fruits from the group of Souri trees with Arbequina nearby and four fruits from the group of Souri plants without Arbequina nearby. Based on five SSR markers, we characterized their pollen donor. The pollen donor of three out of the four fruits sampled from the group of Souri plants with Arbequina nearby, was identified as Arbequina, whereas none of the four sampled from the group of Souri trees located 1.2 km from the nearest Arbequina donor tree had been pollinized by Arbequina.

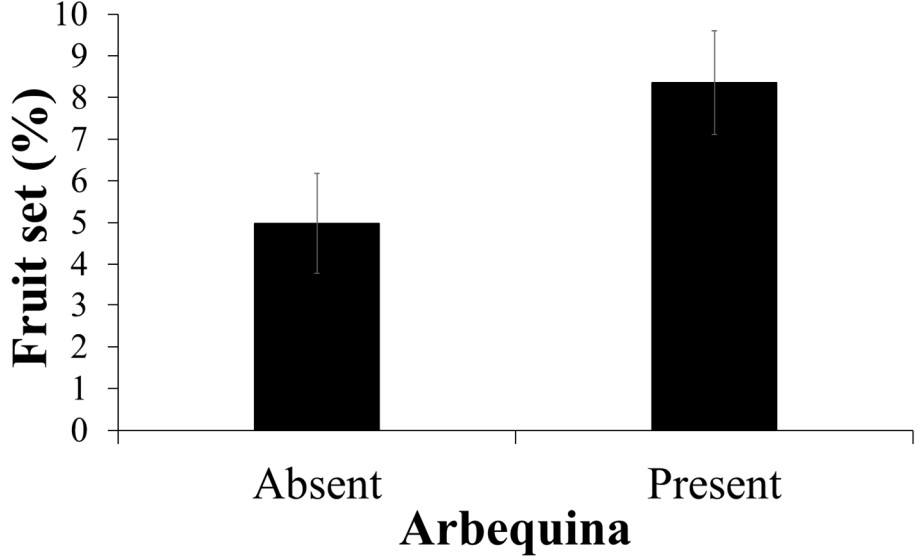

**Figure 4.** Souri fruit set with and without an Arbequina tree in close proximity. Error bars represent confidence intervals ($p < 0.05$).

## 4. Discussion

The Souri is a cultivar highly resistant to drought and to other local edaphic conditions and is the major cultivar in the rain-fed olive groves of Israel. Indeed, the traditional Israeli olive farmers of the Galilee cultivate mono-cultivar orchards of Souri almost exclusively. Since there are many large areas in the northern part of Israel with such orchards with no other olive cultivars nearby, the variety of pollen donors for the Souri in this part of Israel is very limited and the presence of an efficient pollinizer may increase the Souri fruit set significantly, and hence its yield. In this study we identify the Arbequina olive cultivar as the most efficient pollinizer of Souri among several tested cultivars. According to our field trial, adding Arbequina flowering trees to a mono-cultivar Souri orchard will increase fruit set and may increase yield.

For successful cross-pollination to occur, it is necessary to have adequate amounts of compatible pollen available when the flowers are in bloom. This is possible if the compatible cultivars growing in the orchard bloom simultaneously. In our study, the Souri flowering period lasts 17 days during the first half of April and is midway between the early and the late flowering cultivars. Hence, the Souri flowering period overlaps the flowering periods of all tested cultivars. However, for efficient fertilization, overlapping between cultivars effective pollination period (EPP) is needed [1]. EPP is defined as the period when pollination can produce fruit set and depends on optimal pollen viability and stigma receptiveness. This period may last between 4 to 14 days, depending on the cultivar [32].

This period was not elucidated in this study. Another limit in this study is that the flowering period was characterized over one year only and in only one orchard. It is extremely important to characterize the flowering period under different environmental conditions, which can be achieved by characterizing it in different orchards over several years. This will also allow the characterization of the flowering plasticity behavior of the different cultivars [33].

In the experiment we carried out at the Volcani center in 2016, we found the Souri cultivar to be completely self-incompatible. In the same experiment carried out in Deir Hanna, self-pollination of Souri resulted in 0.4% fruit set. However, this fruit set was found only in several branches, whereas most of the branches gave no fruit set at all. Therefore, although it was not verified using DNA markers, we believe that the fruits which developed on these branches are a result of a contamination with foreign pollen which had settled on those particular branches before we covered them with paper bags. It has already been shown that even in mono-cultivar orchards, fruits which developed in self-pollination experiments with paper bags are a result of contamination [22]. Therefor we believe the Souri cultivar to be completely self-incompatible and lacking a mechanism of pseudo-self-compatibility [15].

According to our results in the cross-pollination experiment as well as the paternity analysis, we found the Souri cultivar to be compatible with various other cultivars. These include the local varieties distributed randomly within the Souri orchards, such as Nabali and Amrahani. The cultivars Zakari and Muhasan were shown to be able to pollinate the Souri in our cross-pollination experiment of 2016. However, fruit set of the Souri when pollinated by Zakari and Muhasan was significantly lower compared to that of the fruit set when pollinated by Nabali (Figure 2). In addition, we were not able to identify any Souri embryos with Zakari or Muhasan as the pollen donors by paternity analysis, even though these cultivars are not uncommon among the Souri trees in the traditional mono-cultivar orchards in the northern part of Israel. Therefore, we assume that Zakari and Muhasan are not compatible with Souri. Olive cultivars proved to have a diallelic self-incompatibility (DSI) system which consist of two incompatibility groups (G1 and G2) with a combination of two single-nucleotide polymorphisms that can predict G1 or G2 phenotypes in the olive cultivars [3,34,35]. The olive cultivars Picual and Koroneiki belong to the G2 group, whereas Arbequina, Picholine, and Leccino belong to the G1 group [35]. In a previous study [4] we found that Picual, Souri, and Manzanillo cultivars were compatible with the Barnea cultivar, whereas Arbequina, Picholine, and Koroneiki were incompatible with Barnea. These results together with the results of our current study suggest that the cultivars Souri, Picual, Manzanillo, Muhasan, and Zakari belong to the G2 group and the cultivars Arbequina, Barnea, Picholine, Leccino, Nabali, and Amrahani belong to the G1 group. This is in agreement with the division suggested by Saumitou-Laprade et al. [35]. However, our results regarding the Koroneiki are ambivalent. In the current study, according to the cross-pollination experiment of 2017, the Koroneiki is compatible with Souri (Figure 2). However, in the paternity analysis of the fruit from the Gshur olive grove, we did not find any evidence of the Koroneiki as a pollen donor to the Souri fruits either in 2016, or in 2017 (Figure 3), even though this orchard contains Koroneiki trees in close proximity to the Souri trees. In our previous study [4] we found that the Koroneiki, unlike Picual, Souri, and Manzanilo, is incompatible with Barnea. According to these results, Koroneiki should belong to the G1 group which includes Barnea, Picholine, Arbequina, and Leccino. However, according to Saumitou-Laprade et al. [35], the Koroneiki cultivar was assigned to the G2 group.

Marchese et al. [36] found low levels of inter-compatibility between Arbequina and Koroneiki. This poses a problem if assuming that these cultivars belong to two different group [35]. Possibly, the fact that Koroneiki is a self-compatible cultivar [15,36], may related to its inter-compatibility mechanism behaving differently from that of other olive cultivars.

One of the limitations of this study was that the paternity analysis performed in order to find a better pollinizer than the Nabali, was based on only one multi-cultivar orchard. Therefore, we need to assume that there may be a better pollinizer to the Souri than the Arbequina that was not identified in this study. However, the cross-pollination experiment performed without any human interference (Figure 4), proved that a strategically located Arbequina tree can significantly increase Souri fruit

set. Another limitation was that paternity analysis succeeded only partially (67% in 2016 and 77% in 2017). In the multi-cultivar orchard, which provided the most data of those included in the study, the number of embryos with unidentified pollen donor was negligible (0 and 2, out of 18 and 16, in 2016 and 2017, respectively). In contrast, the number of embryos with an unidentified pollen donor in the mono-cultivar orchards was relatively high. Genotyping several non-Souri trees within the mono-cultivar Souri orchards revealed that these orchards include trees which are not Souri and differ genetically one from the other. We assume that pollen from these random, genetically undefined trees is responsible for the relatively high number of embryos with unidentified pollen donor in the mono-cultivar orchards.

We found that the Nabali cultivar is compatible with the Souri cultivar. However, the Arbequina cultivar was found to be of greater compatibility with the Souri cultivar, in regard to fruit set. This indicates that in addition to the SI groups characterizing the compatibility or incompatibility between cultivars, there exists an additional level of compatibility between pairs of cultivars. In our previous study [4] we also found different levels of compatibility between cultivars, and characterized the Picual as the most efficient pollinizer of the Barnea cultivar. Other studies have also found different levels of compatibility and suggested a rank of efficiency for each compatible cultivar as a pollen donor [20,21]. In the current study, we showed that under field conditions the presence of a flowering Arbequina tree in proximity to Souri trees almost doubled the Souri fruit set. This finding is in agreement with Ayerza and Coates [37] who found that Manzanillo branches that received supplemental pollination in a hot and arid ecosystem, doubled their productivity.

## 5. Conclusions

Mono-cultivar orchards may suffer from a lack of an efficient pollinizer. In an era of climate change, olive fruit set, which is sensitive to environmental conditions [1,2] may be reduced when environmental conditions are harsh. In the current study we demonstrate that compatibility between cultivars in the orchard is critical to achieving maximum fruit set. The effect of compatibility relationships between the various olive cultivars in the field on final oil and fruit yield was not part of the current study, but we assume it will, at least partially, reflect the fruit set level and will increase if compatibility is optimal. The characterization of the incompatibility system in olives as well as the identification of the SI locus [3,15,34,35] is a major step toward an optimal choice of cultivars in planning a multi-cultivar olive orchard. However, the varying efficiency of fertilization between compatible pairs of cultivars, indicates that the optimal pollen donor should be determined for each cultivar. This is true for the planning of new olive groves, as well as for adding optimal pollen donors to existing orchards, especially mono-cultivar orchards. In the future, it would be interesting to determine the yield deficit in mono-cultivar olive orchards when an optimal pollinizer is absent. We believe that carefully choosing the fruit-producing cultivar best suited to local conditions and its optimally compatible pollen donor will result in increased fruit set and higher yields. Hopefully, this will aid farmers in their efforts to reduce the negative effects of climate change.

**Supplementary Materials:** The following are available online at http://www.mdpi.com/2073-4395/10/6/910/s1, Figure S1: Field cross pollination with 'Arbequina' experiment design.

**Author Contributions:** The original design of the study was set up by G.B.-A.; I.B. performed all the molecular analyses; I.B., Y.M., A.M., and G.B.-A. performed the field experiment; I.B. and G.B.-A. performed the data analysis; I.B. and G.B.-A. wrote the manuscript. All authors have read and agreed to the published version of the manuscript.

**Funding:** This work was supported by a grant (No. 20-10-0064) of the Israeli Ministry of Agriculture and Rural Development.

**Acknowledgments:** We thank Yehuda Ben-Ari for valuable assistance in writing and editing this paper.

**Conflicts of Interest:** The authors declare no conflict of interest.

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
