# Peer review of "Compatibility between “Arbequina” and “Souri” Olive Cultivars May Increase Souri Fruit Set†"

_agronomy, doi:10.3390/agronomy10060910_

Round 1
Reviewer 1 Report
Authors showed ‘Arbequina’ was the most frequent pollinizer at Gschur, blooming season of ‘Souri’ and ‘Arbequina’ was overlapped, and ‘Arbequina’ increased fruit set of ‘Souri’. These results suggest that compatibility between ‘Souri’ and ‘Arbequina’olive cultivars may increase ‘Souri’ fruit set.
Regrettably this manuscript has a lot of mistakes and structure need to be improved throughout. Although mistakes in English and format would be improved easily, those in figures and main results are serious problem.
Please see detailed comments below.
Some examples of mistakes in results
Mistakes are, in particular, found in ‘3.2. Artificial cross pollination’. For example, in the legend of Figure 1, Error bars represent…, but no error bar is there. If Figure 1 shows average fruit set, please show a kind of error bars (SD or SE). If not, please show what is this graph.
Furthermore, in the main text, authors wrote that ‘Free pollination gave a fruit set of 3%’. However, Figure 1 shows that fruit set of free pollination was over 4%. Why are these values different? This paragraph includes such kind of confusing values. Please check all results carefully, and provide detailed explanation, data table, and figures. I could not figure out what results should I believe.
Results shown in this MS is too simple to know the real results.
Related with the above comments, if Figure 3 is average, please show SD or SE with row data values. If Figure 3 is not average but total percentage of fruit set, please explain this.
Then in Figure 2, please include number of unknown samples in this figure, since proportion of unknown samples may differ among sites.
Structure of this MS
In ‘Introduction’, lines 73-77, some results and discussions are written like abstract. Furthermore, many discussions are in ‘Result’ section. If this structure would help readers, it would be ok. However, this manuscript seems not to apply this case.
Some examples of minor mistakes
Line 34: Genus name at beginning of a sentence should be spelled out.
Line 34: ‘sporofitic’ may be ‘sporophytic’
Line 64: [19-17,4] should be [4, 14-19]
Line 73: ‘pollinator’ here is strange. It may be ‘pollinizer’.
Line 114: ‘when they…’ should be ‘When they….’
References: some journal names are shown in abbreviations and the others are spelled out.
Reviewer 2 Report
Notes to authors
Hello
I find the general study goal interesting and relevant. The authors have clearly done a lot of fieldwork and various experiments.
The introduction is clearly writing and with some adjustments this would be a very good introduction (see remarks below).
However, the description of the material and methods needs a thorough review. It is confusing to read at the moment. The different experiments are not always clear and explained in sufficient detail in the material and method. The statistics of each analysis are not described for each analysis, this needs to be improved. Also the results section is written quite confusingly.
There is certainly some interesting data in this study. But the description needs to be better and I expect a more logical discussion of the coherence of the different experiments in order to arrive at a clear conclusion.
The lack of replicates in experiment 2.2 is also very important and this limitation should be more prominent in the discussion.
Good luck
General remarks
- Olive is pollinated by wind, how far does pollen of olive travel by wind? How does this impact your study?
- What is the influence of insect pollinators on pollination/fruit set of olive? What is known about this in literature? At least add this to the introduction.
- Did you conduct insect pollinator surveys?
- It’s a pity that the flowering period of the different cultivars was characterized for only 1 year. This limitation needs to be discussed in the discussion
- How long does it take for an olive to ripen? Is 4 weeks long enough? Perhaps in a realistic field setting abortion of fruits still occurs after 4 weeks?
- Why is Barnea chosen for experiment 2.1.2 and not included in experiments 2.1.3 and 2.1.4?
- Statistics are not clear, what are the dependent and independent factors? Which assumptions were checked and what validations were carried out? This is not clear and this is a major limitation of this study at this moment.
- The landscapes of the study orchards are densely packed with olive orchards, this limits the conclusion of the cross-pollination experiment 2.1.4 and 2.2. Landscapes and surrounding orchards need to be described and this limitation needs to be discussed.
- The results from experiments 2.1.3 and 2.1.4 are not reported clearly and it seems like the data of these experiments are analyzed together? This is confusing and not acceptable, split up the analysis, and report the results and the graphical representation and the respective conclusions.
- Only one replicate of the multi cultivar orchard in experiment 2.2, so it is difficult to compare this to four mono cultivar orchards and make strong conclusion. Adjust this in the discussion.
- Experiment 3.4, is it possible to give a description of the surrounding orchards?
Specific remarks
Keywords
I would only include words which are not included in the title
Replace “fruit set” with “fruit production”
Replace “compatibility” with “pollination”
Abstract
Ok, but this needs to be adapted after a thorough revision.
Introduction
Line 30 – “production” instead of “setting”
Lines 41-43 – are these other cultivars distributed sporadically in the Souri orchards? Or in the are they only sporadically present in other orchards in the surrounding landscape? This is not clear to me, rephrase please.
Lines 49-52 – redundant information
Line 55 – in line 34 you indicate that it is about sporofitic, here you mention homomorphic. Which is it?
Line 56 – SI? Shouldn’t this be “DSI”? Otherwise SI should be spelled in full the first time you use a new abbreviation
Line 56 – dominant? Why is S2 dominant or what does this mean?
Line 59 – how is this self-compatibility explained?
Lines 69-72 – what are the disadvantages of SNPs? Also briefly include the advantages and disadvantages of microsatellites. Then state why you choose SNPs over microsatellites.
Material and Methods
Line 81 – start with an indication of the different cultivars which are included in the study (Line 81 or line 84)
Line 93 – when were the branches covered with paper bags? It is confusing that you first state that the flowers were counted and afterwards you say the branches were covered. I would describe this in the opposite order. Or is it possible to count flower before anthesis? If so this needs to be described!
Line 94 – based on what information did you select the Barnea cultivar for this experiment?
Line 98 – how long does it take for an olive to ripen?
Line 100 – include extra spacing between and2018
Line 101 – three different orchards, orchard ID needs to be included as fixed factor in the statistical test, not clear if this is the case…
Line 104 – these 250 flowers, were they already open before the branch was enclosed in the paper bags???
Lines 108-109 – you have 3 orchards and in every orchard 5 trees were selected with 4 branches. There are 5 treatments, how are these treatments distrituted over the different branches/trees???
Line 113 – based on what information did you select the Arbequina cultivar for this experiment? Why not Barnea?
Line 114-115 – strange sentence, rephrase
Line 117 – two different orchards, orchard ID needs to be included as fixed factor in the statistical test, not clear if this is the case…
Lines 117-119 – did you check the surrounding landscapes for other olive trees or for other orchards with other olive cultivars? This might be important? It seems there are a lot of olive orchards in the surrounding landscapes of both orchards… Landscapes and surrounding orchards need to be described and this limitation needs to be discussed.
Line 126 – What do you mean with 7-15 trees?
Line 128-129 – only one replicate of the multi cultivar orchard
Line 134 – remove this subsection header if there is no section 2.2.2
Results
Lines 150-152 – the first two sentences are not appropriate in the results section, move to the discussion
Lines 153-154 – this sentence is not appropriate in the results section, move to the material and methods (cfr remark about line 81 or 84)
Line 164 – Rename this section as it covers more results then only the artificial cross pollination results
Lines 167-168 – which statistical testing was applied? Please show all data?
Lines 170-171 – “sporadically distributed etc….” this is not for the results section but for the discussion or m&m
Line 174 – 3%??? The graph clearly indicates 4,2% for self pollination??
Lines 176-177 – in section 2.1.3 you do not describe any self-pollination treatment???
Figure 1 – please indicate error bars on the barplot.
Line 233 – add the distance between the two orchards already here (1.2 km of line 252)
Experiment 3.4 – is it possible to give a description of the surrounding orchards as this might affect pollination success/fruit set? How many replicates were used in this experiment? I would add the methods part of this section to the material and methods section …
Figure 3 – please indicate error bars on the barplot.
Discussion and conclusion
The main conclusion is based on an observation of one orchard (multi-cultivar) compared to four mono-cultivar orchards. I would like to see a discussion about this limitation in the discussion and conclusion.
Conclusion
It would be interesting to include the determination of pollination deficits in a set of olive orchards in the region to this study or to suggest this as a future research need.
Supplementary
Figure 1 – I would prefer I you would use a colored graph or grayscale graph and indicate the overlap of flowering for each cultivar with the Souri cultivar
- I would include this figure in the main manuscript
References
Check the references, sometimes the journal of publisher (in case of books) are missing (ref 5, ref 6, …)
Round 2
Reviewer 1 Report
I felt that the manuscript was improved. However, I still have comments for methods and results as below.
‘2.1.3. Artificial cross-pollination:’:
How many flowers were examined in each site (S1, S2, and S3) in 2016, 2017, and 2018?
Statistics:
I consider that generalized linear mixed models(GLMM)is better than ANOVA to analyze data in this manuscript. We previously often used ANOVA, and differences of statistical methods would not lead huge differences in main conclusions in some cases. However, lack of necessary information in “2. material & methods” and “3. Results” makes it difficult to judge whether ANOVA can lead correct conclusions. For examples, error bars are not there in Figure 2, F and DF values are not written in statistical results, detailed sample sizes in each treatment or site are not in methods (please see other comments). I recommend to open data online, to reanalyze data, or to add necessarily information and explanation to show that statistical methods in this study have little problem.
Figure 2: Please show error bars on three graphs (2016, 2017, 2018).
‘Paternity analysis’:
In addition to the number of embryos with an unidentified pollen donor, please show the number of measured embryos to suggest the proportion of unknown samples in each site.
Minor comments:
・If order of contents in “2. materials & methods” would consistent with that in “3. Results”, it could be easy to read. ‘Paternity analysis’ and ‘Field cross pollination with ‘Arbequina’’ are now flipped over.
・Please check all of the citation number. For example, [12] is in line 55, and the next citation number is [14] in line 58. [14] should be [13].
・Check all indents. For example, L54 and L111 have no indent, but many of the others have indents.
・Line 165-171. The two paragraphs can be combined because first paragraph has only one sentence.
Reviewer 2 Report
The authors have revised the manuscript and therefore the revised version of the manuscript is an improvement on the first draft.
However, there are still some things which need to improve before this manuscript is ready for publication:
- Emphasize that the paternity analyses only included 1 study location in the abstract.
- Reorganize the material and methods section so that it follows the structure which is described in the abstract as well as the structure in the results section. This will make the manuscript a lot easier and less confusing to read!
My new remarks are highlighted in yellow bellow.
General remarks
- It’s a pity that the flowering period of the different cultivars was characterized for only 1 year. This limitation needs to be discussed in the discussion
Response: We added this limitation to the discussion.- What is the implication of this limitation? Why is it useful to characterize this for multiple years?
See Wenden et al. 2016 A collection of European sweet cherry phenology for assessing climate change
- What is the implication of this limitation? Why is it useful to characterize this for multiple years?
- Why is Barnea chosen for experiment 2.1.2 and not included in experiments 2.1.3 and 2.1.4?
Response: The 'Barnea' chosen for experiment 2.1.2 since it is known to be compatible with 'Souri' (we added in material and methods section the following sentence: "As a control we used 'Barnea' cultivar as a pollen donor, based on our knowledge that 'Barnea' and 'Souri' are compatible"). It was not included in 2.1.3 because in 2016 we only analyzed 'Nabali', 'Zakari' and 'Muhasan' that are sporadically distributed throughout the area of 'Souri' orchards in the north of Israel. In 2017, we had a technical problem collecting 'Barnea' pollen at the Volcani Institute, but had many 'Koroneiki' flowers which are also known to be compatible with 'Souri'. In 2018, we choose 'Arbequina' as our pollen donor, based on the results of paternity analysis.- I would add this explanation to the methods section for the sake of clarity, other readers will also be confused about this.
Specific remarks
Abstract
Line 11: add northern to the first sentence “… in northern Israel.“
Lines 17-19: the paternity analyses was only conducted in one multi cultivar orchard, add preliminary to line 17 to make sure you do not overstate your conclusion, “Based on a preliminary paternity analyses, …”.
Line 25: remove “strategically upwind.” Firstly, this is the first and only time that you mention this and secondly, you did not test this at all, so do not put this in the abstract.
Introduction
Line 35: also add “homomorphic”?
Line 36: remove “compatible”?
Line 43: are these different cultivars present in the wider landscape or do they occur sporadically in orchards? Please specify
Line 57: add a reference for this 12 km dispersal
Line 59: “Olive is characterised …” instead of “It is characterised ..”
Material and Methods
Lines 140-142: “throughout the area” are these different cultivars present in the wider landscape or do they occur sporadically in orchards? Please specify
Lines 108-109 – you have 3 orchards and in every orchard 5 trees were selected with 4 branches. There are 5 treatments, how are these treatments distributed over the different branches/trees???
Response: We added: "In this year, 20 'Souri' branches from each tree were chosen, four branches for each pollen donor."
- How many branches per treatment in years 2017 and 2018?
Line 113 – based on what information did you select the Arbequina cultivar for this experiment? Why not Barnea?
Response: Based on the results of the paternity analysis and artificial cross pollination. However, it is too soon to mention it in the material and methods.
- If that is the case I would re-organize/reshuffle the methods section accordingly:
2.1 Flowering period
2.2 Artificial pollination
2.3 Paternity analysis
2.4 Increasing cross pollination with ‘Arbequina’
The initial 2.1 heading actually does not give much information/structure to the mat & met section. In this way you can mention why Barnea was not used and why Arbequina was used. In addition, both the mat & met section and the results section follow the same structure which will make the paper is easier to read.
Results
Line 174 – 3%??? The graph clearly indicates 4,2% for self pollination??
Response: This was before normalization. We now gave only the normalized values. We also added details of the normalization procedure in the statistical section of "material and methods".
- If data is normalised for the analyses, why don’t you use the normalised data to construct the figure?
Figure 1 – please indicate error bars on the barplot.
Response: There is no need for error bars if there are letters which represent the statistical differences. Therefore, we added the following sentence in the figure legend: "Letters above the columns represent significant differences obtained in Tukey Kramer test (P < 0.05).".
- I still suggest to indicate error bars on the barplot.
Round 3
Reviewer 1 Report
I consider that ‘results’ section is still insufficient. My comments are shown as below.
‘3.3 Paternity analysis’
Lines 241-242: Due to suspected interference by unknown pollen donors and some technical problems.……
What kind of technical problems occurred in paternity analysis?
Unidentified pollen may cause little problem on main conclusion that ‘Arbequina’ was the most frequent pollen donor at Gshur. This is because pollen donors are mostly identified at Gshur. However, in the other areas, proportions of unidentified pollen seem high. If some pollen donors were not distinguished accurately because of technical problems, in sites where unknown samples are dominant, we can not determine main pollen donors.
Please explain why unidentified pollens are not problematic if possible. If unidentified pollen would cause problem in your conclusions, please rewrite conclusions, or add descriptions for the problems. (see below comment for ‘4. Discussion’)
‘3.4 Increasing fruit set by a compatible cultivar’
L.278, please show statistical values. Please check again whether you write enough statistical information in the MS.
‘4. Discussion’
L.325~ Authors discussed about compatibility according to the results in the cross-pollination experiment and the paternity analysis. As I mentioned above, results in the paternity analysis include ambiguity (see above comment for ‘3.3 paternity analysis’). I recommend to discuss about compatibility mainly according to the results in the cross-pollination experiment or to explain enough information about paternity analysis in ‘Results’.
Minor comment
Formatting of statistical results is strange.
(Examples)
Line 180: ‘P(f)=0.987’ should be ‘P=0.987’.
Line 210: ‘P<1.0299X10-11’ should be ‘P=1.0299X10-11’.
Same strange formatting is in ‘3.2 artificial pollination’ and results section.
Reviewer 2 Report
Hello
The manuscript has been improved a lot due to the two review rounds.
I do agree with the other reviewers that the use of GLMM offers more options to perform a more robust statistical analysis. Nowadays the packages are all pretty straightforward to use. GLMM with a binomial distribution for fruit set with appropriate random factors would be my advice here.
I would definitely not give the degrees of freedom in the section2.5 as this is subject of the result section. I suggest to add df as a subscript to the F values, like Fx = ... for instance with x the degrees of freedom.
Good luck
